# How Do Health Care Professionals Perceive a Holistic Care Approach for Geriatric Patients? A Focus Group Study

**DOI:** 10.3390/ijerph20021033

**Published:** 2023-01-06

**Authors:** Denise Wilfling, Jona Budke, Nicole Warkentin, Katja Goetz

**Affiliations:** Institute of Family Medicine, University Hospital Schleswig-Holstein, Campus Lübeck, 23538 Lübeck, Germany

**Keywords:** geriatric patients, care and case management, holistic care, qualitative study

## Abstract

Background: Geriatric patients require holistic care in order to meet their complex care needs. The project RubiN (Continuous Care in a Regional Network) provides case and care management (CCM) for older people to address these needs in a primary care setting in Germany. This study aimed to explore the experiences of health care professionals who provided CCM for geriatric patients. Methods: Focus group interviews with general practitioners (GPs), health care assistants (HCAs), and case managers (CMs) were conducted. Transcribed data were analyzed by using qualitative content analysis. Results: Ten focus group discussions (*n* = 15 GPs, *n* = 14 HCAs, *n* = 17 CMs) were conducted. The different health care professionals emphasized the importance of a holistic care approach to geriatric care. Moreover, the GPs stated that the CMs supported the patients in organizing their care. A CCM could help encourage patients to remain at their own homes, which would have an effect on patients’ quality of life and satisfaction. Conclusion: A well-functioning and effective cooperation between those health professionals involved is a prerequisite for a trustful relationship in the holistic care of older people. This creates a feeling of security for all people involved in the care process.

## 1. Introduction

As the population ages, the proportion of geriatric patients in the healthcare system is rising [1]. Due to multimorbidity in geriatric patients, this patient group has complex care needs, which could be challenging for health professionals [2]. Especially general practitioners are challenged by geriatric patients, which is compounded by the fact that older people desire to stay in their own homes for as long as possible [2,3]. In order to meet these complex care needs, integrated and coordinated care within a community is recommended as it helps to reduce hospitalization, rehospitalization, as well as institutionalization [4]. Care and case management (CCM) is an important component of integrated care and requires cooperation between different professions in a community and could be seen as a holistic care approach [5,6]. The German Association of Care and Case Management defines care management as a systems-related procedure and Case Management as a case-related procedure [7]. Moreover, the Case Management Society of America defines case management as “a collaborative process of assessment, planning, facilitation, care coordination, evaluation, and advocacy for options and services to meet an individual’s and family’s comprehensive health needs through communication and available resources to promote patient safety, quality of care, and cost-effective outcomes” [8].

In general, it is recommended to combine different interventions for the assessment, planning, coordination, monitoring, and evaluation of care [9]. 

There have been several projects dealing with case management based on the conditions in Germany [10], but the implementation of case management in the German health care system is still a slow progress. The Advisory Council on the Assessment of Developments in the Health Care Sector (SVR) recommended the involvement of non-medical staff in the care of patients [11]. Based on the results of previous projects and the SVR recommendations, the new case management intervention RubiN (Regional ununterbrochen betreut im Netz; Continuous Care in a Regional Network) was developed. 

Several studies have shown that CCM can help to meet the complex care needs of geriatric patients [12,13,14]. This results from the fact that CCM comprises a variety of interventions, focusing on health care needs, the coordination of required care, as well as the consideration of social aspects [9,15]. Effects of CCM on patient-related outcomes have been evaluated in several studies, but they have mainly focused on hospitalization, lengths of stay, hospital readmission, and mortality [4,16,17]. Several studies have shown that trust is a key element for the successful homecare of geriatric patients [9,18]. When patients have a specific contact person they trust, and health professionals involved in care trust each other, this has positive effects on the care situation and care outcomes [19]. It is necessary to provide person-centered care based on individual needs in order to gain the trust of the patients. Person-centered care is considered the best practice care for geriatric patients with multiple conditions and requires interprofessional collaboration due to different health care providers and the involvement of family caregivers [20,21]. A few studies have shown that the satisfaction of patients and their quality of life has improved, which was mainly reported by the patients themselves [12,22,23]. However, there is a lack of studies assessing the impact on patients’ health and well-being experienced by different perspectives, especially by health care providers [9]. Therefore, the aim of this qualitative study was to explore the experiences, especially the perceived changes of health care professionals, with the offered CCM for geriatric patient care in the project RubiN. 

## 2. The Materials and Methods 

### 2.1. The Project RubiN

The project RubiN was developed to provide regional CCM for outpatient care of older people (age > 70 years) in a primary care setting in Germany. Five practice networks implemented a CCM for their geriatric patients living at home. The patients received a CCM provided by the care and case manager (CM). The CM assessed, coordinated, and monitored the home care of the geriatric patients in consultation with the general practitioner (GP) who is, besides the CM, a part of this practice network. The health care assistants (HCA) work in the GP’s practice and support the CM as well GP in the coordination of patient care. The aim of this multi-professional, cross-sectional, and assessment-based CCM was to reduce gaps in care and to enable older patients to retain as long as possible at their own homes. Detailed information about the project and the intervention was published in Gloystein et al. [24]. Moreover, after 12 months, an extensive process evaluation was conducted. The acceptance and feasibility of the intervention were evaluated by means of case analyses, focus group meetings, and expert interviews [24].

### 2.2. Study Design

This study addressed the experiences of health care professionals who provided CCM for geriatric patients and was designed as a qualitative study, based on the COREQ (Consolidated criteria for Reporting Qualitative Research) guideline [25]. A qualitative study design allows for a deeper understanding of individual participants’ viewpoints [26]. Focus groups enable an open discussion by sharing experiences with a group of people [27]. Therefore, focus group interviews with health professionals (GPs, HCAs, CMs) were conducted.

### 2.3. Study Population and Recruitment

The study population consisted of health professionals from the medical practice networks participating in the project RubiN. Detailed information about RubiN has been published elsewhere [8]. Participants in the focus group interviews were recruited from June to July 2020 by the project coordinators of each medical practice network. After agreeing to participate, one of the authors (NW) contacted the project coordinators for the exact date arrangement. The project coordinators should consider that participants in each focus group interview should be heterogeneous, including GPs, HCAs, and CMs because of the multi-professional character of the case management intervention. Additionally, heterogeneous focus groups seem to be more effective to discover different perspectives and ideas due to different skills and knowledge of participants [28,29].

All participants received information about the background, procedure, and aim of the study. Informed consent was obtained by a signed consent form in advance of participation. Each participant was remunerated 50 Euros after participation.

### 2.4. Data Collection

Focus group interviews were conducted between August and September 2020. The focus groups were carried out in the five practice networks which were located in different regions (western, northern, and eastern) in Germany. An equal distribution of the number of focus groups and a sufficient number of health care providers who were responsible for geriatric care within the five practice networks were desired. Overall, ten focus group interviews with three to seven participants were conducted. All focus group interviews were conducted by the same female researchers (KG (sociologist and health services researcher), NW (physician)) to ensure a constant and homogeneous interview process. The focus group interviews were recorded digitally, fully transcribed verbatim, and anonymized. All transcripts were in German and their accuracy was validated by comparing the transcripts with the digitally recorded interviews. 

For the focus group interviews, a semi-structured topic guide was developed by an interprofessional team of physicians and health researchers, in consultation with representatives from the medical practice networks (managing directors, project coordinators). The guidance contained questions about experiences with interprofessional coordination in the context of interprofessional care for geriatric patients as well as perceived changes for patients as a result of such a care model. The guidance was piloted during the first focus group interview. Afterward, single formulations and the order of the questions were modified. Since the results of the first focus group did not differ significantly from others, the first focus group was also included in the evaluation. For more information on the guide see Appendix A.

### 2.5. Data Analysis

Data were analyzed according to the qualitative content analysis, [27] using the computer software Atlas.ti. 8.4. [30]. The research team used a deductive–inductive approach in generating categories. Categories and subcategories were generated deductively based on the interview guide. The developed category system was adjusted inductively during the analyzing process according to the meaning of transcripts. New categories were added in line with an inductive approach. Transcripts were coded independently by two researchers (DW (nursing background) and JB (physician background)) into main- and subcategories. Data were compared regularly in consensus meetings with intensive discussions, supported by a third researcher (KG). As a result of this, detailed documentation of the development of the category system was provided, which allows traceability of the whole research process, and fulfills an essential criterion for good qualitative research [31]. Detailed information about the analyzing process is given in Appendix B.

### 2.6. Ethical Approval

Ethics approval was obtained from the ethics committee of the University of Luebeck in Germany in July 2019 (Approval number: 19-282). No additional data were evaluated. Informed consent was obtained through a signed consent form and included the publication of anonymized responses.

## 3. Results

### 3.1. Sample Characteristics

Overall, ten focus groups with *n* = 46 participants (*n* = 15 GPs, *n* = 14 HCAs, *n* = 17 CMs) were conducted. The duration of the focus groups varied and was on average 75.6 min (±13.5). The characteristics of the study participants are shown in Table 1.

### 3.2. Key Categories

The key category “perceived changes in care” included different subcategories, namely “insight into patient’s living environment”, “promoting home care”, “quality of life”, “patient satisfaction”, and “feeling of security”.

### 3.3. Subcategory: Insight into Patient’s Living Environment

This subcategory describes how insight into the living environment of the patients has been changed through CCM. Home visits conducted by the CMs provided the opportunity to get insights into a patient’s living environment. The CMs got impressions of the living situation including existing smells, impressions of the food supply, and medication compliance. Interprofessional care enabled a holistic view of the patient. Therefore, care needs could be identified, which would otherwise have been unnoticed.


*“In addition, there was a second mainstay of what we could do from a medical perspective. There was somebody who could take a look into the homes. A lot of patients, who might need home visits, or if we say it would make sense, we should go to their home and take a look, we don’t have that time. We have restrictions and we cannot do that. At least, I think that was a big gain from this project, that we got insights in patient’s living environment”.*
(RubiN_FG6_GP)

In some cases, desolate living conditions were uncovered that had previously remained hidden behind facades.


*“There are many who get dressed up again for the doctor’s office and everything is great and you sit at the doctor’s office and ‘No, no, I’m fine and everything is fine.’ And we come in at home and we see: Okay, it’s not that fine anymore, is it?”*
(RubiN_FG5_CM)

However, sometimes only small things were identified, but contributed to better care and encouraged patients to remain at home. Besides GPs and CMs, HCAs were also surprised by how many aspects were detected by home visits conducted by the CMs.


*“You also get to know how the patients are doing, what they are doing and things we didn’t know about”.*
(RubiN_FG3_HCA)

### 3.4. Subcategory: Promoting Home Care

The results of this subcategory describe how care at home was promoted by a CCM. Through the insight into the patient’s living environment, it was possible to plan interventions and initiate support options in order to encourage patients to remain at their own homes. Next to medical and therapeutic problems, social aspects were considered. Interventions to promote living at home included adjustments at home to achieve accessibility, the use of aides, consulting home care services, and the organization of services to help with shopping, cleaning the house, or ordering meals on wheels.


*“One patient lives alone […] He fell again and again, always came to the clinic. I also made two house visits. The flat was chaotic. It’s not surprising that he fell. Then he only had one cleaning lady who came once a week and by the time we had organized something, he was back in the clinic […] Then I registered him in the RubiN project, then all-round care was organized and since then he has no longer been in the clinic […] now a home help comes and a care service comes once a day and he gets good food and doesn’t eat the moldy things from the fridge and so on”.*
(RubiN_FG4_CM)

The organization and coordination of all these interventions allowed patients to live safely in their own home.


*“I make sure that people are cared for at home and are allowed to stay at home to avoid inpatient stays. I believe, that to date, I only had a few patients who were moved to an institution, I think 7 out of over 200 patients. And that’s not a lot”.*
(RubiN_FG1_CM)

The doctors also wished that they did not have to admit their patients to hospital so quickly and were happy about the integration of the patients into their familiar environments.

### 3.5. Subcategory: Quality of Life

The subcategory “quality of life” describes the increase in patient quality of life. An increase in patient quality of life was perceived both by GPs and the CMs. By taking social aspects into account, patients appeared happier and showed an increased zest for life. The promotion and retention of independence had a positive effect on participation in activities. The patients participated more in sports groups, made new friendships, and went on trips, e.g., to parks.


*“And when you see how happy the patients are and what steps they make, in certain areas, simply because they participate more, then that’s totally awesome”.*
(RubiN_FG1_CM)

An improvement in quality of life was especially observed by the CM. Due to their regular home visits, different problems were detected and could be solved, as the following example showed:


*“I also have a 94-year-old patient who lived upstairs for a year, practically locked up on her 1st floor in the apartment, could not go down because she could no longer walk the stairs. And we got her physiotherapy and all sorts of things, together with the general practitioner and now she can walk the stairs again. And that’s easy… And was able to go to a park again and watch her flowers and was super happy”.*
(RubiN_FG6_CM)

In addition, it was observed that the socio-medical wishes were often not expressed in the doctor’s office and that medical values such as the INR value also stabilized because of the CCM.

### 3.6. Subcategory: Patient Satisfaction

Results in this subcategory describe how satisfied patients were with CCM. All health professionals perceived their patients to be satisfied with the health care model. The patients themselves recognized that they benefit from the project and were satisfied with their positive developments as well as negative ones. They were especially satisfied with organized aids, living room adaptations, and support with applications. Having the opportunity to get in touch with someone and talk about their own well-being or problems had a positive effect on the well-being of patients.


*“The conversation, well to talk about things that may never have found a place and space. Suddenly, they come to light. And that’s also a form of well-being for the patient. Generally, we want self-determination and stability. And when you are free from circling thoughts, you suddenly feel better. Or to talk to clear their conscience in order to try and change things”.*
(RubiN_FG2_CM)

By participating in the care model, patients were reached who might otherwise have fallen through the cracks and their need for care would not have been uncovered otherwise.

Through the care model, patients were freed from fears and burdens by giving them the feeling that someone was taking them by the hand. At the same time, many new possibilities to try something new were shown to the patients to get some variety in everyday life, especially for patients who require very little support currently.


*“Some patients were just happy, that something new–they were in their daily grind, doesn’t give a lot of variety, or–and we also founded a sports group, and some of them said “Great! Just coming out again, getting new input, someone is coming home”.*
(RubiN_FG4_CM)

Moreover, the HCAs in the GP practice received positive feedback and perceived that the patients were happy and satisfied with the provided care. This led to an easier and friendlier treatment in the doctor’s office.


*“And then we often get really positive feedback from the patients. That’s nice, it’s going well”.*
(RubiN_FG9_HCA)

### 3.7. Subcategory: Feeling of Security

This subcategory describes how all participants developed a feeling of security through CCM. Both GPs and CMs reported that patients gained a strong feeling of security through regular contact with CMs, as they had a fixed contact person who could be reached at any time and was trustworthy.


*“That’s now a certainty, when you can pour out your heart to someone, and find a trustful level. That’s a feeling of security, if there is someone you can call every time, who is listening to you, is coming to you at home. So that is a very big advantage, I think we have other ways to talk. And that would probably slip through our fingers because we do not go at home and maybe she is not coming to us. Or she is coming, but then everything is a little different. And until we realize that there is a problem, it sometimes takes a little bit longer”.*
(RubiN_FG6_GP)

Since many older people have no relatives or they often live far away, home care created a feeling of security, which is very important from a psychosocial point of view and reduced their fears.


*“Many single older patients were simply less lonely, less psychologically burdened because of the suggestions of leisure activities, that really was–is just a great help”.*
(RubiN_FG3_HCA)

Doctors perceived that the patients felt safer when a person who works closely with the GPs regularly came to a patient’s home, which finally also gave the doctors a feeling of security.


*“So it is very important for me as a GP that there is also an additional security, of course. That I know that a patient is not only cared for by me, but also by a geriatric coordinator […] because the care of patients is becoming more and more complex, in general. And the demands are increasing overall. And that is also a great help”.*
(RubiN_FG5_GP)

CMs felt secure because they had a clearly defined work and care task for which they were optimally supported by good cooperation in the interprofessional team.


*“So I think it’s nice to work in a team and not to say on your own: Now decide […] but you can exchange ideas together and I think that’s really an enrichment in terms of content, professionally and also mentally, a relief to say: ‘No, we decided together like this, we’ll try it again at home now”.*
(RubiN_FG7_CM)

## 4. Discussion

The purpose of this present qualitative study was to explore health professionals’ perspectives on the effects of the newly developed care model on patient-related outcomes. The main findings of this study show that a CCM can help to encourage patients to remain at their own homes, which has a noticeable effect on patients’ quality of life and satisfaction. This corresponds to the personal wishes and ideas of many older people, namely living at home for as long as possible [3]. An interview study with older people underlines the importance of a CM for their care that resulted in feelings of security at home for these people [32].

It was shown that interprofessional care allows us to see patients from a holistic perspective. This holistic perspective also includes an insight into the individual living situation and into the living environment of each patient. This enables health professionals to plan interventions at an early stage, to encourage patients to remain at home, and finally to avoid hospital admissions or institutionalization. Additionally, having someone get in touch with the patient and talk about their own sensitivities and concerns showed positive effects on the patient’s well-being. These results are in line with the results of other studies evaluating the impact of integrated care on patient-related outcomes [9,32,33,34].

Geriatric patients feel a sense of security through the care provided by CMs. In turn, this sense of security has positive effects on mental health, and patients become more empowered [32]. Psychosocial support was seen as essential for a trusting relationship and for the patient to finally gain a feeling of security in their ability to be able to cope with future concerns and challenges [9,33,34]. For this reason, CMs must have various skills in order to meet these requirements. Different findings underlined that CMs are seen as highly competent individuals with a variety of skills, medical skills, and case management skills to provide continuity of care on the one hand, and “soft” skills to provide social support on the other hand [32,33].

Also, CMs see this as one of their main tasks in integrated care and enjoyed their new role, having more time to listen to patients’ individual needs [35]. Even family caregivers perceived these extensive competencies of CMs, especially their soft skills, as had already been reported in a previous study [36]. Moreover, further studies are important to quantify our results of the focus group, such as the evaluation of patient-related outcomes by patients who received specific care with a reliable and valid tool. The patient perspective is an additional important indicator for quality of health care [37]. Moreover, the offered CCM in RubiN can be seen under the aspect of social, prescribing a way to promote health and wellbeing for older people, which should be considered in further studies [38]. Several projects dealing with case management play a crucial role in supporting patients who are navigating complex health care systems [10]. Moreover, a systematic review shows that case management appears to be a promising method to support patients facing complex care situations especially if assessment, care planning, coordination, monitoring, and health education will be addressed [39]. 

A qualitative study design was chosen to answer the underlying research question. However, the results cannot be generalized due to the qualitative study design. Only people who were directly involved in the RubiN project took part in the present study. The project coordinators of the different practice networks performed the recruitment. Therefore, it is unknown how many GPs, HCAs, and CMs were contacted for participation in the focus group. The presented results are therefore subject to a selection bias. Moreover, the participants were very positive in their statements; thereby, a social response behavior could not be excluded. The heterogeneous composition of the focus groups could implicate power issues related to hierarchy. However, the moderator of the groups encouraged participants to feel safe and be open in their statements. Additionally, participants of the RubiN project were remunerated 50 Euros. Furthermore, it cannot be ruled out that only those people who were extremely motivated or satisfied with the RubiN project took part in the focus groups.

## 5. Conclusions

The results of this study are essential for the development and implementation of integrated geriatric care and allow for a deeper understanding of community-based CCM interventions. Geriatric patients can benefit from interprofessional care. Well-functioning and effective cooperation between those involved health professionals is a prerequisite for a trustful relationship between health professionals as well as between CMs and patients. This creates a feeling of security in all persons involved in the care process and is also reflected in geriatric patients.

## Figures and Tables

**Table 1 ijerph-20-01033-t001:** Characteristics of study participants (*n* = 46).

Characteristics	
Age in Years, Mean (Standard Deviation)	46.6 (± 20.5)
Gender, Female, *n* (%)	37 (80.4)
Function in the Medical Practice Network, *n*	
General Practitioners	15
Health Care Assistants	14
Case Managers	17
Years of Working in the Medical Practice Network, *n* ^$^	
Less Than One Year	1
One to Five Years	13
Six to Ten Years	8
More Than 10 Years	7

^$^ varies due to missing values.

## Data Availability

The datasets generated during the current study are not publicly available due to the ethical requirements but are available from the corresponding author upon reasonable request.

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
