# Peer review of "How Do Health Care Professionals Perceive a Holistic Care Approach for Geriatric Patients? A Focus Group Study"

_ijerph, 2023, doi:10.3390/ijerph20021033_

Round 1
Reviewer 1 Report (Previous Reviewer 2)
Dear authors,
Thank you for a great review of the manuscript! The results satisfactory improved, but the analysis could be developed more. I also notice that all the results are positive. Were there any "negative" findings at all? Discuss in limitations?
Reviewer 2 Report (New Reviewer)
The paper is a good effort in documenting HC professionals' viewpoint on holistic care approach for geriatric patients. The main weakness lies in not providing background to Holistic care for such patients and what it actually means in the medical field. In general, the manuscript has an improved content after addressing comments from another reviewer or editor. Another shortcoming is the very short conclusion section with no outlook on future research or policy as well as limitations of this work, if any.
This manuscript is a resubmission of an earlier submission. The following is a list of the peer review reports and author responses from that submission.
Round 1
Reviewer 1 Report
Even though the topic is extremely relevant to the society
The intervention Rubin Project is not properly integrated in the literature. What other models exist? What support is found in the literature about the relevance of this model vs others?
The discussion is also poor and the contribution is obvious (like someone who is selling the Rubin project) and not fully justified and theoretically integrated the literature
Additionaly in the method, and being a qualitative study
Researchers role in regards to the analysis made is missing
Further details about the recruitment process are missing
Reviewer 2 Report
General: Don not use the term "elderly" as it may be stigmatic. Use in stead "older people" or "older person" to increase person centerdness.
Please define Case manager and Care and case manager in the introduction, as these functions may not exist or is named something else in different countries.
Introduction: It is a bit confusing as the text in lines 38-46 seem to be an description of your study context and should be in the methods section. If not, you should clarify this.
It is a somehow confusing about the context of the study as in the introduction is not clear if you focus on hospital care or home care. Please clarify
The literature review is a bit shallow. Aren't there any more studies in this area?
Methods: I think you need to say some more about the RubiN-project as the context isn't clear. Se previous comment. What was the focus of the project? settings?
Did you mix the different professions in each focus group? Please clarify. If you did or if you didn't, discuss in the method discussion section
The reason of including the different professions isn't clear. Nor in the methods or in the introduction.
Analysis: You say: "The research team used a deductive-inductive approach in generating categories. Categories and sub-categories were generated deductively based on the interview guide". What was the interview guide?
What kind on content analysis did you do? You need to describe each step very clear for the reader to be able to judge the trustworthiness of the analysis and thus of your study.
Also: "The developed category system was adjusted inductively during the analysing process." What do you mean? how, why?
"Transcripts were coded independently by two researchers (DW [nursing background], JB [physician background]) into main- and sub-categories". what do you mean by that? there is a lot of steps to go from transcripts to main and sub-categories.
"As a result of this, a detailed documentation of the development 112 of the category system was provided,". Where can I fin that? please provide a table or appendix for the reader to see.
I also wonder how wise it is to mix participants due to professions. The professions commonly have different perspectives.
Results: The aim: "to explore the experiences especially the perceived changes of health care professionals with the offered CCM for geriatric patient care in the project RubiN ". First: the project RubiN needs to be described to be able to understand your study and thus your results. Otherwise, the aim are understandable. You also say that you focus on changes. What was the starting point? Do you have a baseline to compare with? If not, you may not write it in a way that it has increased or that the project has effected an so on..
The sub-categories are very short and as almost half of them are quotations. I think the analysis is not quite done yet. Some of the subcategories could be collapsed into similar.
Discussion: it is clear and informative, but it needs to be revised based on previous comments of the results, methods and introduction
Limitations: Please see comments on the method and revise according to that.
What influence on the trustworthiness may it be that the participants got 50 euro?
